Corrected: Publisher correction

# Dispersal homogenizes communities via immigration even at low rates in a simplified synthetic bacterial metacommunity

Stilianos Fodelianakis [1], Alexander Lorz[2], Adriana Valenzuela-Cuevas[1], Alan Barozzi [1], Jenny Marie Booth[1] & Daniele Daffonchio [1]

Selection and dispersal are ecological processes that have contrasting roles in the assembly of communities. Variable selection diversifies and strong dispersal homogenizes them. However, we do not know whether dispersal homogenizes communities directly via immigration or indirectly via weakening selection across habitats due to physical transfer of material, e.g., water mixing in aquatic ecosystems. Here we examine how dispersal homogenizes a simplified synthetic bacterial metacommunity, using a sequencing-independent approach based on flow cytometry and mathematical modeling. We show that dispersal homogenizes the metacommunity via immigration, not via weakening selection, and even when immigration is four times slower than growth. This finding challenges the current view that dispersal homogenizes communities only at high rates and explains why communities are homogeneous at small spatial scales. It also offers a benchmark for sequence-based studies in natural microbial communities where immigration rates can be inferred solely by using neutral models.

[1] Biological and Environmental Sciences and Engineering Division (BESE), Red Sea Research Center, King Abdullah University of Science and Technology (KAUST), Thuwal 23955-6900, Saudi Arabia. [2] Computer Electrical and Mathematical Science and Engineering Division (CEMSE), King Abdullah University of Science and Technology (KAUST), Thuwal 23955-6900, Saudi Arabia. These authors contributed equally: Stilianos Fodelianakis, Alexander Lorz, Adriana Valenzuela-Cuevas. Correspondence and requests for materials should be addressed to S.F. (email: stelios.fodelianakis@kaust.edu.sa) or to D.D. (email: daniele.daffonchio@kaust.edu.sa)

Environmental selection and dispersal[1] are key assembly processes for microbial communities[2,3] and their quantification could help us to predict how microbial communities will change in the future[4]. Microbial communities tend to diversify when selection is variable, i.e., when different taxa occupy different niches along environmental gradients. Typical examples include pH gradients in soils[5] or temperature gradients in the oceans[6]. In contrast, communities tend to homogenize when dispersal is high[7], for example, in permeable sediments[8] or at small scales in soil[9].

Disentangling the contrasting roles of selection and dispersal is particularly important for metacommunities (a set of communities that are linked by dispersal[7]) in which dispersal can decrease the strength of selection. This is the case for many ecosystems in freshwater and marine habitats (i.e., oceans, lakes, rivers, and streams), the pore water in sediments of water bodies and streams, estuarine systems with deltas, etc. In such ecosystems, because dispersal for microbes is passive at macroscopic scales[10,11] (albeit active dispersal like chemotaxis[12] operating at microscopic scales), transfer of material is positively correlated to immigration. This can homogenize environments across a metacommunity, weakening selection. Thus dispersal could homogenize a metacommunity indirectly and only when it weakens selection (Fig. 1a). Alternatively, dispersal could homogenize a metacommunity directly via the immigration of individuals without selection being weakened (Fig. 1b). In the latter case, selection is strong and the taxa in the different communities are growing differently, but immigration of individuals due to dispersal homogenizes their populations along the metacommunity. In this scenario, the metacommunity would be homogenized only when a certain ratio between growth and immigration of individuals is reached. To investigate such mechanisms regarding the role of selection and dispersal, targeted experiments, rather than static observations, are required[13].

Here we investigate the mechanism by which dispersal homogenizes a metacommunity where dispersal and selection are coupled. We use a synthetic bacterial community (Fig. 2a) and a sequencing-independent monitoring strategy to quantify the absolute abundances of each bacterial population along the metacommunity (Fig. 2b). We deploy a controlled experimental set-up using a closed circulation system where the medium is flowing among three incubation vessels, each one set at a different temperature (Fig. 2c). We follow the dynamics of the metacommunity in this system under varying degrees of dispersal and temperature-driven selection in five experiments, each one performed at a different circulation speed (Fig. 2d). We first grow the communities without dispersal, imposing only variable selection because the growth of each strain varies depending on temperature (Fig. 2e). We then grow the metacommunity at four different circulation speeds. This causes dispersal to increase and selection to decrease because the temperature difference among the three incubation vessels decreases (Fig. 2d) and that should decrease the differences in the growth of each strain among the vessels.

Finally, we simulate the heat transfer and population dynamics of the system in 100 scenarios of intermediate dispersal to pinpoint exactly at which circulation speed, and how, dispersal effectively homogenizes the metacommunity. We find that dispersal homogenizes the metacommunity directly via immigration and that happens without selection being weakened. Homogenization occurs not only when dispersal is strong, as theoretically expected[7,14], but also when it is weak, i.e., even when immigration is four times slower than growth.

## Results

**Creating a traceable synthetic bacterial metacommunity.** We first aimed to build up a synthetic community composed of easily detectable strains. To that end, we isolated 317 bacterial strains from soil (Supplementary Table 1, Supplementary Figure 1) and we screened their forward and side-scattering profiles with staining-free flow cytometry. We discriminated three isolates (namely, B42, E111, and E310) whose scattering profiles overlap minimally during the lag and the exponential phases of growth (Supplementary Table 2, Supplementary Figure 2, Supplementary Movies 1–3), indicating that their populations could be easily discriminated in a mixed culture. The distinct profiles of the isolates correspond to their different morphologies (Fig. 2a and Supplementary Figure 3). We drew non-overlapping gates in the front/side scatter (FSC-A/SSC-A) plots that contained the majority of the events of samples from pure cultures (Fig. 2b—top

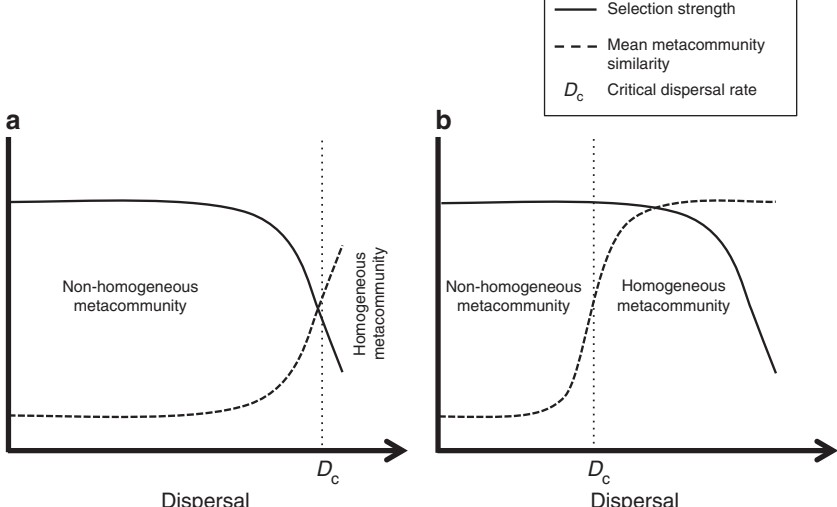

**Fig. 1** Proposed ecological mechanisms by which dispersal can homogenize a metacommunity. **a** Dispersal homogenizes a metacommunity indirectly, by weakening selection across habitats via transfer of material, and only after selection is weakened. In this case, the critical dispersal rate required for homogenization ($D_c$) is high. **b** Dispersal homogenizes a metacommunity directly via immigration of individuals and the metacommunity is homogenized before selection is weakened. In this case, $D_c$ is lower than in scenario A and depends on the ratio between growth and immigration of individuals. In both scenarios, $D_c$ will depend on how strongly selection and dispersal are coupled

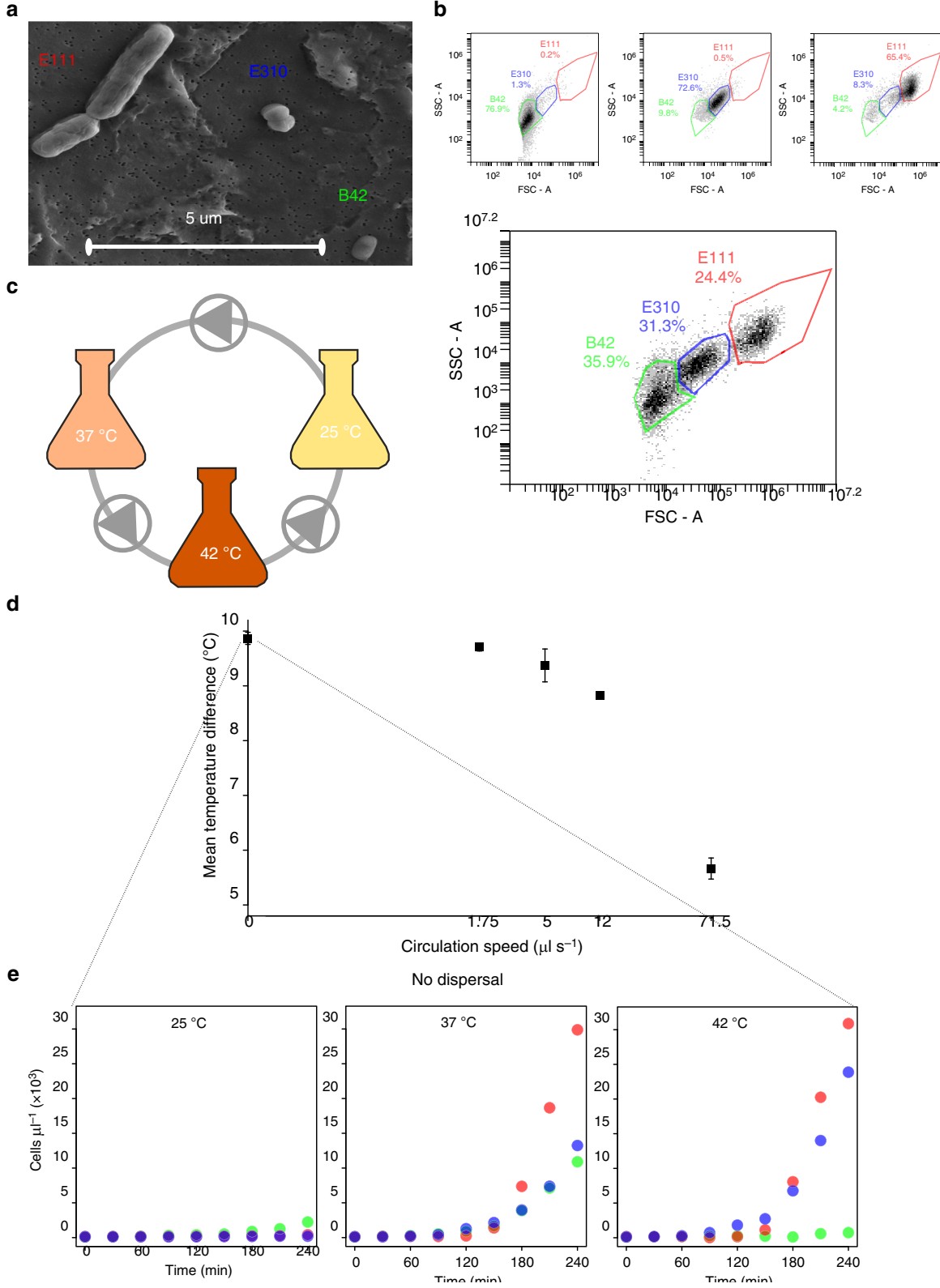

and Supplementary Table 3). We defined these gates as "representative" and we used them to quantify the total populations of each strain in the synthetic community, correcting for overlapping of one population to the other two "representative" gates (Fig. 2b—bottom and Supplementary Tables 4-6). We monitored the growing populations with a temporal resolution of 30 min and with a high accuracy (average accuracy of 92–99.6%; Supplementary Table 7).

We then created a metacommunity of three initially identical synthetic communities that were physically connected (Fig. 2c), and we grew this metacommunity under variable selection and increasingly strong dispersal (Fig. 2d). We applied variable selection by incubating the three synthetic communities at 25, 37, and 42 °C (nominal). This creates different community dynamics at different temperatures (Fig. 2e) because the growth of the three strains depends strongly on temperature irrespectively of the

**Fig. 2** Overview of the synthetic community and experimental approaches. **a** Scanning electron microscopic (SEM) image of the three bacterial members of the synthetic community. E111: *Bacillus* sp., E310: *Staphylococcus* sp., B42: *Chryseobacterium* sp. **b** Forward scatter area (FSC-A—$x$ axis)/side scatter area (SSC-A—$y$ axis) plots of pure cultures (top) and of a mixture of the three bacteria (bottom). Colored lines represent non-overlapping gates that we drew based on pure cultures to calculate each population in the mixed culture. We show all three gates for the pure cultures (top) as examples for the calculation of spillover ratios (Methods). **c** Overview of the experimental set-up. Flasks represent incubation vessels at different temperatures (nominal values indicated on the flasks), gray curves represent tubing, and triangles within circles represent peristaltic pumps and the direction of the flow. All three bacterial strains were present within each incubation vessel at a 1:1:1 initial ratio, and all pumps had the same flow direction and speed. **d** The mean temperature difference among the three incubation vessels in the experiments with increasing dispersal. The $x$ axis is in logarithmic scale and therefore zero is put at the beginning of the axis conventionally for illustration purposes (the real value at that point is 0.05 $\mu$l s$^{-1}$). Error bars correspond to one standard error ($n = 3$). Source data are provided as a Source Data file. **e** The growth of each strain in the community over 4 h at each incubation vessel without dispersal. Each point represents the average of three replicate samples with an average coefficient of variation of 4.1%. We excluded the error bars to assist visualization. Source data are provided as a Source Data file

---

stage of growth (Supplementary Figure 4A). We monitored the first 4 h of incubation in order to capture the temperature-dependent differences in the duration of the lag phase and in the early exponential phase among the strains (Fig. 2e, Supplementary Figure 4B). We recorded significant mutualistic interactions among the three strains during this 4-h incubation at all temperatures; the growth of the strains was promoted in the mixed cultures compared to the pure cultures at all temperatures (analysis of covariance (ANCOVA), $n = 9$, $0.041 > p > 3.95E-06$, Supplementary Data 1). We then placed the incubation vessels within a closed circulation system where medium is flowing at a specific direction and with a constant flow via peristaltic pumps (Fig. 2c), creating a metacommunity of three connected communities. We applied increasingly strong dispersal to this metacommunity by growing it at four different circulation speeds (Fig. 2d).

**Laboratory experiments with increasing dispersal**. We next examined at which circulation speed the metacommunity was effectively homogenized by defining two metrics that are based on the Bray–Curtis (BC) similarity and that are calculated for each circulation speed: "BC within" and "BC across" (Fig. 3a). "BC within" is the mean pairwise similarity within the metacommunity and quantifies how similar the communities among the three incubation vessels are to each other (Fig. 3a, solid two-headed arrows). "BC across", on the other hand, quantifies how much the communities change compared to when there is no dispersal (Fig. 3a, dashed two-headed arrows). Both metrics compare communities from the same time point, i.e., every 30 min, starting at the 60th min of incubation because before that communities are very similar due to lack of growth (Supplementary Figure 5). If at a given circulation speed "BC within" is significantly higher than "BC across", we call the metacommunity homogeneous because the communities are more similar among them than when there is no dispersal.

We found that the metacommunity is homogeneous in the experiments with circulation speeds of 5, 12 and 71.5 $\mu$l s$^{-1}$ (Fig. 3b). At zero circulation speed, "BC within" is 46.97% and it gradually increases to 60.51, 78.26, 89.92, and 97.96% at circulation speeds of 1.75, 5, 12, and 71.5 $\mu$l s$^{-1}$, respectively. At the same time, "BC across" is 73.72% at a circulation speed of 1.75 $\mu$l s$^{-1}$ (it cannot be defined for zero dispersal) and it decreases to 59.75, 53.67, and 54.13% at circulation speeds of 5, 12, and 71.5 $\mu$l s$^{-1}$, respectively. The increase in the standard deviation while both indices decrease reflects the increasing differences in the compared communities with increasing incubation time (Supplementary Figure 5). The metacommunity is non-homogeneous at 1.75 $\mu$l s$^{-1}$ ("BC across" > "BC within", linear mixed-effects model, $n = 63$, $t = -3.194$, two-sided $p = 0.009$) and becomes homogeneous at circulation speeds of

$\geq 5$ $\mu$l s$^{-1}$ ("BC within" > "BC across", linear mixed-effects model, $n = 63$, $9.75 < t < 13.37$, two-sided $p < 0.001$) (Fig. 3b). This indicates that the metacommunity becomes homogeneous at a circulation speed between 1.75 and 5 $\mu$l s$^{-1}$.

**Simulations with increasing dispersal**. To pinpoint exactly at which circulation speed the metacommunity becomes homogeneous, we simulated the dynamics of the metacommunity under 100 scenarios of intermediate circulation speeds using an Ordinary Differential Equation (ODE) modeling framework. We split the range of circulation speeds logarithmically, sampling more densely at low speeds, because our experimental results suggested that the transition point was between 1.75 and 5 $\mu$l s$^{-1}$. In our ODE framework, the 4-h incubation at each scenario is split into very small time steps where three processes occur: (1) the temperature in each community changes according to heat gain or loss (from the vessel, the ambient environment, and the transported medium), (2) the populations of each strain in each community grow depending on temperature, and (3) cells are transferred passively from community to community according to the circulation speed and the direction of the flow. We modeled the changes in temperature based on energy equilibrium (Fig. 4a), the growth of each strain with a lag phase followed by an exponential phase, with both phases depending on the strain and on temperature (Fig. 4b–d), and the transfer of cells depending on the volume of the transferred medium (for details, see Methods). Our modeling framework replicates accurately the growth dynamics of the metacommunity in the experiments with dispersal ($0.93 < R^2 < 0.999$, Supplementary Figure 6, Supplementary Table 8), and the resulting modeled communities are 84–98% similar (in terms of BC similarity) compared to the actual communities from the respective experiment.

Using the output of the ODE modeling, we calculated "BC within" and "BC across" for all the simulations with intermediate dispersal and we found that the metacommunity becomes homogeneous at a circulation speed of 3.85 $\mu$l s$^{-1}$ (generalized linear mixed-effects model, $n = 21$, $z = 4.46$, $p = 0.0016$; Fig. 5a). More specifically, we found that "BC within" was significantly lower than "BC across" from 0 to 2.17 $\mu$l s$^{-1}$ (generalized linear mixed-effects models, $n = 21$, $-21.45 \leq z \leq -3.54$, $0.0001 < p \leq 0.04$), indicating that at this range of circulation speed the metacommunity was still significantly heterogeneous. After that, we observed a transition zone from 2.23 to 3.55 $\mu$l s$^{-1}$ where "BC within" was not significantly different from "BC across" (generalized linear mixed-effects models, $n = 21$, $-3.28 \leq z \leq 3.05$, $0.1 \leq p \leq 0.23$). From 3.85 $\mu$l s$^{-1}$ and on, "BC within" was significantly higher than "BC across" indicating that the metacommunity is homogenized (generalized linear mixed-effects models, $n = 21$, $4.46 \leq z \leq 18.73$, $0.0001 < p \leq 0.0016$).

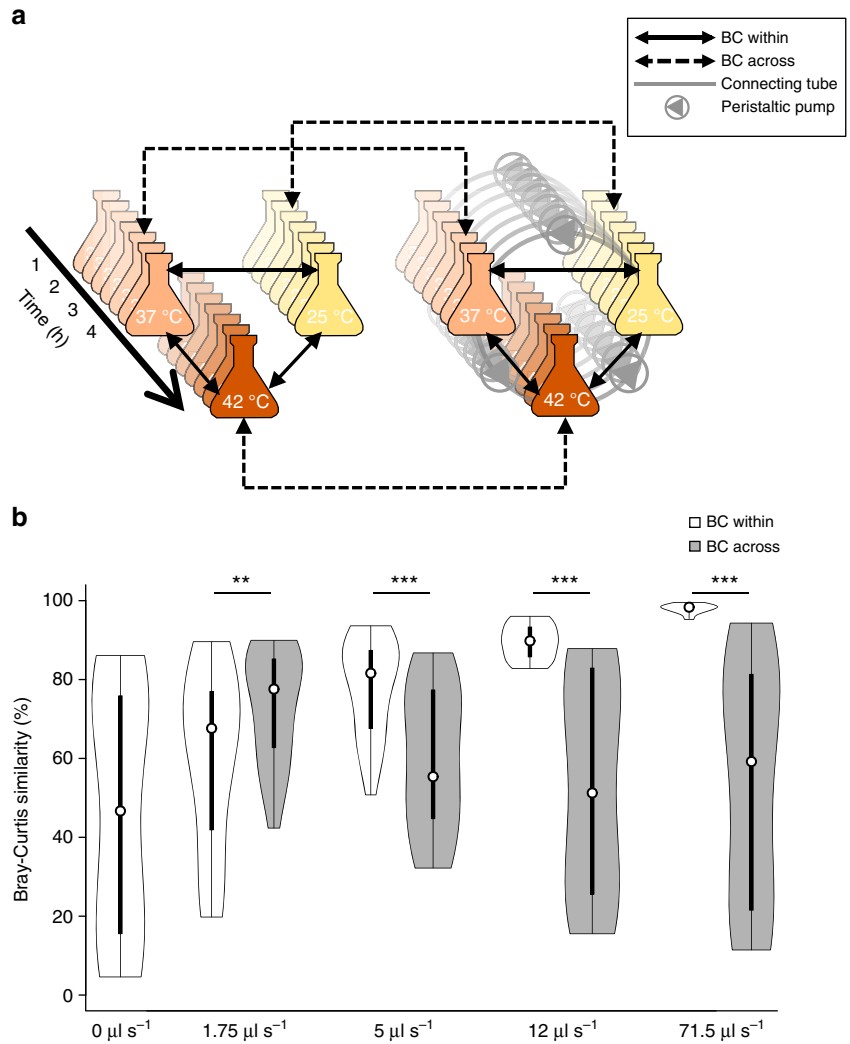

**Fig. 3** Calculation method and changes in Bray–Curtis (BC) within and BC across. **a** The communities from the experiment without dispersal are shown on the left side of the panel and the communities from an experiment at a given circulation speed are shown on the right side. "BC within" compares communities from the same experiment and time point (and thus at different temperatures), whereas "BC across" compares communities at the same temperature and time point (and thus from different experiments). For each experiment performed at a given circulation speed, each metric is calculated as the average of three comparisons per time point (starting after the first hour of incubation) for a total of 63 comparisons over three replicate experiments. **b** Black hollow circles within the boxes correspond to median values, vertical rectangles to 25–75% boxplots, and vertical lines to the range of the remaining observations. The sides of the boxes correspond to the kernel density of the observations. Asterisks correspond to $p$ values of linear mixed-effects models comparing "BC within" to "BC across" at each tested circulation speed ($n = 63$, double asterisks (**) and triple asterisks (***) for corrected two-sided $p$ between 0.01 and 0.001 and $p < 0.001$, respectively). Source data are provided as a Source Data file

**Examining how the metacommunity is homogenized**. We further examined whether dispersal homogenizes the metacommunity indirectly via weakening selection or directly via immigration of cells and we found that it does so via the latter (Fig. 5b). We used the variability (coefficient of variation (CV)) in the growth rates of each strain among the incubation vessels within the metacommunity as a proxy for selection, because it expresses how differently the strains grow at the different vessels. We used the growth-over-immigration ratio to examine how fast the populations within the vessels are replaced by immigrant cells at a given circulation speed. We found that the growth-over-immigration ratio decreases fast, while the CVs of the growth rates are stable (for strain E310), increase slightly (for strain E111), or decrease slightly (for strain B42) at circulation speeds of <5 µl s$^{-1}$ (Fig. 5b). At the transition point of 3.85 µl s$^{-1}$ where the metacommunity is homogenized, the growth-over-immigration ratio decreased 283-fold compared to the simulation at the lowest circulation speed

(0.05 µl s$^{-1}$). On the contrary, the mean temperature difference decreased by 0.5 °C (Supplementary Figure 7), resulting in a 56% decrease in the CV of the growth rates for strain B42 (the strain with the lowest growth overall—Fig. 4b, Supplementary Figures 6, 11), no changes for strain E310, and an 11% increase for strain E111 (Fig. 5b, Supplementary Data 2). This indicates that, at the transition point, the strains are still growing very differently within each incubation vessel but immigration is sufficiently high to homogenize the metacommunity. Surprisingly, the growth-over-immigration ratio at the transition point is 4.187, which indicates that the average growth rate is more than four times higher than the average immigration rate.

To further examine the role of immigration and heat coupling in homogenizing the metacommunity, we performed simulations under two scenarios in which: (a) the vessels are thermally coupled but there is no migration of cells across vessels and (b) there is migration of cells but no thermal coupling across

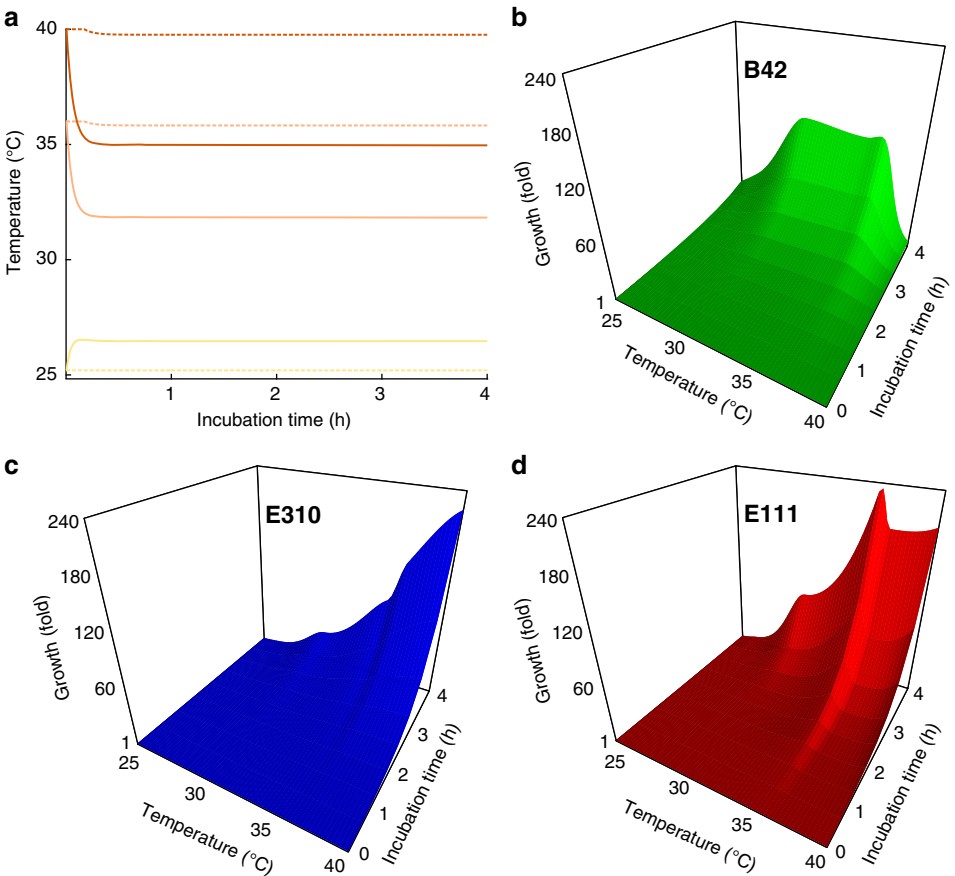

**Fig. 4** The modeled changes in temperature and the growth of each strain over time. **a** The dashed and the solid curves represent the temperature changes at the lowest (0.057 μl s$^{-1}$) and the highest (71.5 μl s$^{-1}$) circulation speed, respectively. Temperature values for intermediate circulation speeds fall between these two lines. Yellow represents vessels at nominal incubation temperature of 25 °C, light orange vessels at 37 °C, and dark orange vessels at 42 °C. **b–d** Temperature is given on the x axis, time is given on the y axis, and growth is given on the z axis. In all panels, temperature corresponds to the actual temperature within the incubation vessels, not the nominal

vessels. We then compared the community dynamics from these simulations to the original model (Fig. 6a) at a range of circulation speeds up to 5.72 μl s$^{-1}$ where the metacommunity is already homogenized in reality (Figs. 3b and 5). We found that the metacommunity is homogenized very similarly to the original model without thermal coupling (Fig. 6c) but not without immigration (Fig. 6b) where "BC within" and "BC across" do not converge and the metacommunity is non-homogeneous at the examined range of circulation speeds. The transition zone in the scenario without thermal coupling is between 2.4 and 3.55 μl s$^{-1}$ and the community becomes homogenized again at 3.85 μl s$^{-1}$ (generalized linear mixed-effects model, $n = 21$, $z = 3.83$, $p = 0.0129$) where the growth-over-immigration ratio is 4.5.

Finally, we tested the generality of our findings by simulating the entire system two more times and we found similar results at both simulations (Supplementary Figures 8–9). In these simulations, we dissected a circulation speed gradient from 0 to 200 μl s$^{-1}$, and we caused dispersal to weaken selection faster by lowering ~3 times the heat-buffering capacity of the incubation vessels. We also relaxed the initial strength of selection by making the growth profiles of the strains more similar than in reality along the temperature gradient (Supplementary Figures 8A–9A). The latter caused the starting value of "BC within" to be higher than in reality in both of the simulations; 57.8 and 58.9% compared to 46.97% in reality (Supplementary Figures 8B–9B). Despite these differences, we found, overall, similar results in both simulations compared to the main experiment. Dispersal

homogenizes the metacommunity not via weakening selection but via immigration, and at the transition point the growth rate is at least two times faster than the immigration rate (Supplementary Figures 8C–9C).

## Discussion

In this study, we examined the roles of immigration and selection in a synthetic bacterial metacommunity that is analogous to natural ecosystems where immigration and the strength of selection are inversely coupled. Our experimental system represents the general case of a metacommunity under strong variable selection whose strength can be gradually alleviated by increasing dispersal. Analogous natural systems include all those habitats where bacteria can disperse passively via the flow of water; practically, this means all aquatic systems where metacommunity theory is applicable[15].

Our results show that dispersal homogenizes communities via immigration even at low rates, thus providing insights into open microbial ecology questions. Conceptually, dispersal is expected to become homogenizing only at high rates[7,8,14] or not homogenizing at all[13], and observations in nature have supported both cases[16–20]. However, in the latter studies only few dispersal conditions were tested and the communities were screened qualitatively. Moreover, our study concerns changes in community structure (species abundances), not in community composition (presence/absence of species—in our study, all strains were

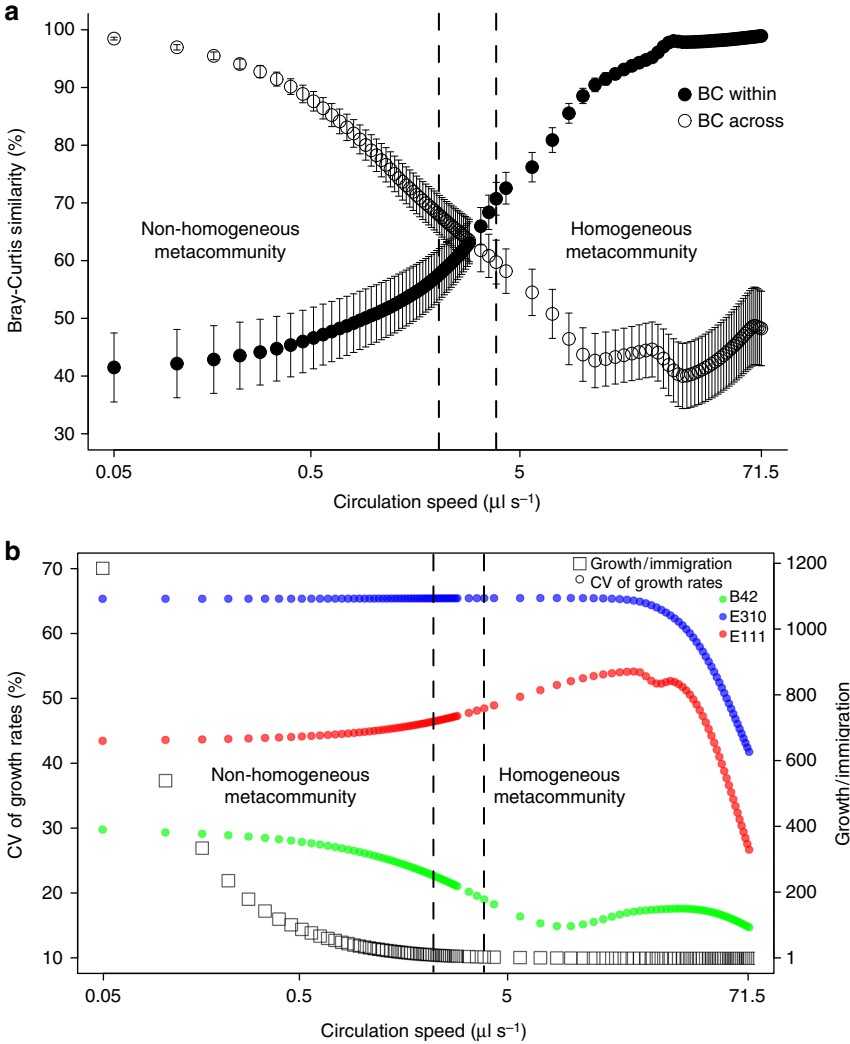

**Fig. 5** The modeled changes in community dynamics with increasing dispersal. **a** Bray–Curtis similarity as a function of increasing circulation speed (in logarithmic scale). Vertical dashed lines delimit the range of dispersal under which "BC within" is significantly lower than "BC across" (on the left of the left line) and vice versa (on the right of the right line; generalized linear mixed-effects models, corrected two-sided $p < 0.05$). Vertical bars represent one standard error ($n = 21$). Source data are provided as a Source Data file. **b** The coefficient of variation (CV) of the growth rates among the three incubation vessels (left side) and the growth-over-immigration ratio (right side) as a function of increasing circulation speed (in logarithmic scale). The vertical dashed lines are drawn based on **a**

initially present in all communities at a 1:1:1 ratio), whereas other studies focused on both compositional and structural changes. Differences in community composition are less likely to occur at small spatial scales, because at these scales the distribution of microorganisms is rarely found to be dispersal-limited[10]. This can explain why homogenizing dispersal is evident at relatively small spatial scales[8,9,14,21]; at these scales, differences in communities are structural rather than compositional and according to our results such communities can be homogenized at low immigration rates compared to growth.

Our findings also offer a benchmark for sequence-based studies in natural microbial communities where researchers often use models like that of Sloan et al.[22] to examine where a metacommunity is neutrally assembled (and therefore homogenous). Our findings suggest that a metacommunity can be homogenized even when the probability that a new individual within a community arrives from the metacommunity rather than from within the community itself is less than one out of five. This corresponds to an $m$ parameter of the neutral model of Sloan et al.[22] of <0.2. Thus $m$ parameter values of ≥0.2 in natural surveys may suggest

that homogenizing dispersal is prevalent in the examined metacommunity. However, complementary approaches such as the null modeling framework of Stegen et al.[8] should also be applied to such studies to support that notion.

Furthermore, our flow cytometry-based method offers a significant advantage over conventional phylogenetic screening. Our method captures changes in the absolute population densities of each community member accurately, whereas conventional phylogenetic screening[14,23,24] captures changes in relative abundances[25,26] (which can also be distorted[27]) and requires large sampling efforts[26]. Assessing β-diversity based on changes in the absolute abundances of taxa can reveal patterns that are undetectable with conventional screening[28]. This is because changes in relative abundance can reveal how a taxon's dominance within a community, but not its proliferation, changes[29]. For example, in two communities consisted of three species with populations of 100, 100, 100 and 200, 200, 200 cells, respectively, the BC similarity based on relative abundance would be 100%, whereas the same metric based on absolute abundances would be 33.33%. In this example, the use of relative abundances overlooks the fact

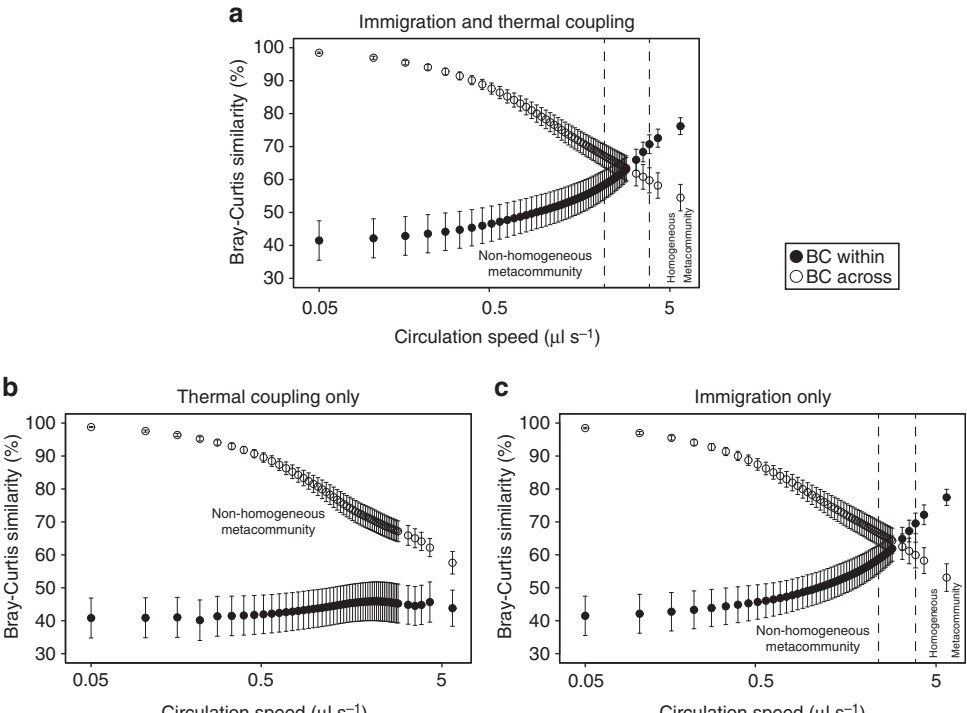

**Fig. 6** Community dynamics under the original model and in two alternative scenarios. **a** Dynamics in the original model that includes both immigration and thermal coupling. **b** Dynamics in the model that includes only thermal coupling. **c** Dynamics in the model that includes only immigration. Bray–Curtis (BC) similarity (y axis) is plotted as a function of increasing circulation speed (x axis, in logarithmic scale). Vertical dashed lines delimit the range of dispersal under which "BC within" is significantly lower than "BC across" (on the left of the left line) and vice versa (on the right of the right line; generalized linear mixed-effects models, corrected two-sided p < 0.05). Vertical bars represent one standard error (n = 21). Source data are provided as a Source Data file

that in the second community all species have double the populations compared to the first community. Furthermore, absolute population counts allow for precise mathematical modeling through ODE, as we applied in this study, or through individual-based modeling[30].

Moreover, our framework can be applied to a broad range of isolates in combination with the recent method of Rubbens et al.[31]. Here we used manual gating and bacterial strains that have very distinct front/side scattering profiles at all the different growth stages, because these profiles served as an "internal control" to verify the validity of our gating strategy. However, this can limit the number and morphological diversity of the suitable strains. In future studies, isolates with less distinct profiles can be used and their populations can be classified with the supervised learning algorithm of Rubbens et al.[31] on any combination of parameters that may also include fluorescence (and thus naturally fluorescent strains).

Overall, our study is based on a simple synthetic meta-community and mathematical modeling and thus comes with some important limitations. Even though synthetic communities and controlled in vitro experiments offer ideal systems to test ecological hypotheses[32], they cannot reproduce the complexity of natural systems. For example, in our study we impose selection by only one abiotic factor (temperature), whereas in nature selection is imposed by a wide range of co-occurring and interacting biotic and abiotic factors. However, in principle, the homogenization of the physicochemical parameters within a meta-community due to the physical transfer of material (including not only carbon sources and nutrients but also bacterial predators) should relax multiple selection factors simultaneously. Another limitation of our monitoring method is the limited number of taxa in our synthetic community; more complex patterns could have emerged if we repeated the same experiment with more taxa

(even though we obtained the same results by changing twice the growth characteristics of our taxa in silico). Future works that investigate more diverse synthetic communities and different kinds of abiotic and biotic selection like limiting resources or predator–prey dynamics would give further insights to the interaction between variable selection and homogenizing dispersal. In addition, in our experiments we started from a perfectly even metacommunity, but since community evenness influences community functionality[33,34] we cannot exclude that it could also affect our experimental outcome. For example, the growth of the strains at different temperatures could differ if we started from a metacommunity that was not even because of the significant interactions among the strains. Finally, our findings regarding the importance of migration and weakening of selection can be further verified experimentally using set-ups where migration and selection can be uncoupled. For example, selection could be isolated by circulating sterile medium so that there is no migration, and migration can be isolated by starting from communities of varying composition and evenness under identical environmental conditions. Despite its limitations, our study highlights the potential of dispersal to homogenize microbial metacommunities and paves the way for similar future studies to tackle questions that are otherwise very hard to answer in natural systems with sequence-based approaches.

## Methods

**Isolation and identification of strains**. We isolated bacteria from three samples (~100 g each) of an arid soil at King Abdullah University of Science and Technology (22.308442 N, 39.107747 E) to find strains that form discrete populations using fluorescent-independent flow cytometry. We plated soil suspensions diluted at $10^{-6}$ on two types of agar media, R2A soil extract[35] and Luria Bertani (LB) (Sigma-Aldrich), and we incubated the plates at 30 °C for a minimum of 48 h. After incubation, we picked single colonies from the plates and streaked them in new plates three consecutive times to ensure the purity of the isolated strains. We

selected colonies based both on their physical characteristics and randomly. We then screened the selected isolates phylogenetically by sequencing of the full-length 16S rRNA gene by performing two PCR reactions for each isolate: one reaction using primers 27F (5'-AGAGTTTGATCATGGCTCAG-3')[36] and 806R (5'-GGAC TACHVGGGTATCTAAT-3')[37] and the other one reaction using primers 515F (5'-GTGCCAGCMGCCGCGGTAA-3')[37] and 1492R (5'-TACGGTTACCTTGTTAC-GAC-3')[38]. The amplification conditions were as follows: 2 min initial denaturation at 95 °C followed by 30 cycles of denaturation at 95 °C for 30 s, annealing at 54 and 52 °C for 30 s for the first and the second reaction, respectively, and elongation at 72 °C for 1 min. We performed each 20 µl reaction using 1 unit of native Taq DNA polymerase (Thermo-Fisher Scientific), 0.5 µM of each primer, 1× of the respective reaction buffer, 1.5 mM MgCl$_2$, 0.25 mM dNTPs, and 1 µl of template DNA. We prepared the template DNA fresh by diluting colonies of each isolate into 50 µl of Tris-HCl buffer (10 mM, pH = 8) and incubating each dilution for 15 min at 95 °C. We clustered the 16S rRNA gene sequences of each isolate into operational taxonomic units (OTUs) of ≥95% of sequence similarity, broadly corresponding to different bacterial genera[39], using the UCLUST[40] algorithm. Finally, we identified the taxonomy of one representative strain from each OTU using the online BLAST platform and comparing against the "refseq_rRNA" database of NCBI (https://blast.ncbi.nlm.nih.gov/Blast.cgi).

**Screening of isolates with flow cytometry.** We screened liquid cultures of one representative isolate per OTU using a BD Accuri C6 flow cytometer (Becton Dickinson) to find isolates that form distinct populations on a FSC (x axis)/SSC (y axis) biplot. Our aim was to use isolates with distinct scattering profiles to assemble a synthetic community in which we could trace the populations of the different strains using fluorescent-independent flow cytometry.

For each representative isolate, we diluted overnight cultures, which were prepared by inoculating a single isolate colony into 5 ml of LB medium, in filtered (through a 0.2-µm syringe filter, Corning, Germany) and sterilized physiological solution (NaCl 0.9% w/v) to a final dilution of $10^{-1}$–$10^{-3}$ depending on the strain. We analyzed aliquots of the final dilution at "Slow" acquisition speed (14 µl min$^{-1}$) for 2 min. We set the threshold of recording an event at 10,000 regarding the height signal of the FSC. Using these settings we recorded at least 50,000 events for each strain, while the filtered and sterile physiological solution had negligible background noise, i.e., 10–20 events per µl that were also widely scattered (Supplementary Figure 10—left side). We plotted the events on FSC area (x axis)/SSC area (y axis) biplots with the axes at a logarithmic scale. Finally, we calculated the mean values and the CV for FSC and SSC area signals of each isolate (Supplementary Table 2).

**Scanning electron microscopy of bacterial cultures.** We filtered pure and mixed liquid cultures of strains B42, E310, and E111 onto 0.1 µm polycarbonate Whatman filters (Nucleopore) before fixation in 2.5% glutaraldehyde in 0.1 M cacodylate buffer for 72 h. After washing in 0.1 M cacodylate buffer, we post-fixed the samples in osmium tetroxide. We then rinsed the samples in sterile distilled water, washed them with an ethanol gradient (from 20% to 100%) and subjected them to critical point drying (Autosamdri-815B, Tousimis). We attached the filters to aluminium stubs with carbon tape and coated them with a 5-nm layer of Au/Pb using a K575X sputter coater (Quorum). We acquired the images using a Quanta 600 FEI (Thermo Scientific) scanning electron microscope at an acceleration voltage of 5 kV.

**Representative populations, spillover, and growth rates.** We grew the three selected strains in pure cultures to find the "representative" gates, i.e., non-overlapping two-dimensional areas in the FSC-A/SSC-A plots that contain the majority of the total recorded events of each pure culture at a given time, and the spillover ratios, i.e., the proportion of the events of a pure culture that fall in the "representative" gates of the other two pure cultures at a given time. The spillover ratios were expressed in relation to the "representative" populations. As an example, we show the FSC-A/SSC-A profiles of each pure culture at the beginning of the incubation on the upper side of Fig. 2b. The "representative" gate for the pure culture of B42 (green color, top left plot) contains 76.9% of the total recorded events, while 1.3% of the total events of the same sample fall in the gate of E310; therefore, at this time point, the spillover ratio of B42 to the gate of E310 is 0.0169. Likewise, the spillover ratio of E310 to the gates of B42 and E111 is 0.135 and 0.007, respectively (blue color, top middle plot), and the spillover ratio of E111 to the gates of B42 and E310 is 0.127 and 0.064, respectively (red color, top right plot). At least 50,000 events were recorded for the calculation of the "representative" gates and the spillover ratios, using a threshold of 10,000 regarding the height of the FSC signal.

We incubated the cultures of the three strains at 25, 37, and 42 °C for 4 h; the temperatures and duration were chosen to correspond to the temperature gradient and to the duration of the circulation experiments, respectively. We sampled, drew the "representative" gates, and calculated the spillover ratios at the beginning of the incubation and at every 30 min until the end of the fourth hour of incubation (Supplementary Tables 4–6). We used the "representative" gates and the spillover ratios of pure cultures to find the individual populations of each strain in mixed cultures. To that end, we solved the following system of equations for

each time point:

$$B42_r = B42_o - E310_r \times S_{E310/B42} - E111_r \times S_{E111/B42} \qquad (1)$$

$$E310_r = E310_o - B42_r \times S_{B42/E310} - E111_r \times S_{E111/E310} \qquad (2)$$

$$E111_r = E111_o - E310_r \times S_{E310/E111} \qquad (3)$$

where $B42_r$, $E310_r$, and $E111_r$ are the real "representative" populations of B42, E310, and E111, respectively, $B42_o$, $E310_o$, and $E111_o$ the observed events in the gates of B42, E310, and E111, respectively, at the same time point, and $S_{a/b}$ the spillover ratio of strain "a" to the "representative" gate of strain "b". Since the spillover of E111 to the "representative" gate of E111 was negligible (<0.0025 at all time points), the respective term was omitted from Eq. 3. Then we found the total individual populations of each strain by considering the percentage of total events of each isolate that fell within the corresponding "representative" gate at each temperature, based on the profiles of samples of pure cultures (Supplementary Table 3). To assess the accuracy of the method, we performed tests where we measured the population densities of the three strains in pure cultures and then mixed 100 µl of each pure culture and calculated the population densities in the mixed culture. We compared the expected population densities from the pure cultures to the observed ones from the mixed cultures after tripling the latter to account for the three-fold dilution due to mixing (Supplementary Table 7).

For the incubation, we placed 30 ml of sterile LB medium (Sigma-Aldrich) in 50 ml falcon tubes at each temperature (in triplicate) and we inoculated each one with overnight cultures of all three strains at a starting cell density of 453–473 cells µl$^{-1}$ (~150 cells µl$^{-1}$ of each strain). The recorded events in the sterile LB (Supplementary Figure 10—right side) were deducted from the total counts for each gate prior to any downstream calculation. We incubated the selected strains at 25, 37, and 42 °C (corresponding to the temperature gradient of the experiment), and we calculated the population densities in each community at every 30 min for 4 h of incubation.

To examine potential interactions among strains, we compared the growth rates in the mixed cultures versus the growth rates in monocultures at 25, 37, and 42 °C (Supplementary Data 1). For that, we grew the strains in monocultures as described above, starting both from population densities comparable to the total starting population densities in the mixed growth assays and from population densities comparable to the per strain starting population densities in the mixed growth assays.

**Set-up of the circulation experiment.** We set up a system where LB medium was circulated among three falcon tubes (50 ml capacity) with the aid of a 3-head Masterflex® FH 100 M peristaltic pump (ThermoFisher Scientific, USA). We placed each of the falcon tubes, henceforth called "incubation vessels", within a Thermomixer (Comfort model, Eppendorf, Germany) that we set at a constant nominal temperature: one at 25 °C, one at 37 °C and one at 42 °C. All three incubation vessels were shaken at a constant speed of 300 rpm to facilitate homogeneous mixing. We drilled the caps of the falcon tubes to insert plastic tubing for the entrance and exit of medium to and from the falcon. We performed four individual experiments, each one replicated three independent times, where the speed of the peristaltic pump was set at 1.75, 5, 12, and 71.5 µl s$^{-1}$ and the flow of LB was from 25 to 37 to 42 °C and back to 25 °C (Fig. 1c). All falcon tubes, tubing, caps, and valves were ethanol-sterilized before the experiment, and the whole system was placed under a laminar flow hood to ensure sterile conditions.

On the day before each experiment, we placed 30 ml of sterile LB within each incubation vessel, keeping the flow off, to ensure that the medium reaches temperature equilibrium within the incubation vessel. The recorded starting temperatures before the beginning of each experiment were 25, 35.5, and 40 °C for the medium within thermomixers at nominal temperatures of 25, 37, and 42 °C, respectively.

At the beginning of the experiment, we inoculated each incubation vessel with a starting population density of 120 ± 2.8 cells µl$^{-1}$ for each strain. To start with as similar as possible communities, we prepared a single inoculum by mixing overnight pure cultures at a 1:1:1 cell ratio and we inoculated the same volume from that inoculum to each incubation vessel. We collected and analyzed triplicate samples of 333 µl from each vessel immediately after the inoculation before turning on the flow and at every 30 min after that until the end of the fourth hour of incubation, for a total of 9 samples per time point.

**Experimental data acquisition and analysis.** We analyzed the culture samples during the experiments with a BD Accuri C6 flow cytometer (Becton Dickinson) using the "low" acquisition speed, i.e., 14 µl min$^{-1}$ and, for 90 s per sample, recording events above a 10,000 threshold at the FSC-H. This flow cytometer comes with standard factory settings for the voltage of the lasers that cannot be changed by the user. We gated the individual populations of the three strains based on the scattering profiles of the pure cultures, also considering the shifts in the populations through time (Supplementary Movies 1–3) and correcting for spillover (Supplementary Tables 4–6). To determine the instrument variability, we measured 6 different mixed culture samples 12 times each and we found that the average CV in the mean values of the counts in each gate was 3.75%. To test whether our

sampling in "Slow" speed setting yields significantly different results compared to the minimum recommended settings according to the instructions of the instrument's manufacturer, we performed additional measurements of all the three pure cultures in both settings. We used overnight cultures, we made the appropriate dilutions, and we split the diluted cultures in six subsamples of 1 ml per strain to create as identical samples as possible. We placed all the subsamples on ice and we measured three of them, per strain, with "Slow" settings and three of them, per strain, with the "minimum recommended" settings. We drew new gates for the subsamples at "minimum recommended" settings because the increase in the flow speed compared to "Slow" settings causes the events to shift slightly upwards in the FSC-A/SSC-A biplots and the populations of the different strains to be resolved less well (Supplementary Figure 11). We sampled volumetrically 20 μl per sample with both settings. We found that there were no significant differences between the recorded events at the different settings (t tests, n = 3, p > 0.05 at all cases), with the counts using the "minimum recommended" settings being consistently 3–5% lower on average than the counts using the "Slow" settings. This difference is, however, indistinguishable from the recorded background noise of the instrument (CV = 3.75%).

**Modeling changes in temperature at intermediate dispersal**. To model how temperature changes in scenarios with dispersal, we started from the energy conservation principle, which dictates that the change in energy in each tube, $\Delta E_{\text{tube}}$, is equal to the sum of the energy transferred from the thermomixer, $E_{\text{thermomixer}}$, and the energy of the LB medium being pumped in and out, $E_{\text{pumped in/out}}$. To this end, we use the following energy balance equation for each of three tubes:

$$E_{\text{thermomixer}} + E_{\text{pumped in/out}} = \Delta E_{\text{tube}} \qquad (4)$$

When the experimental time course $t$ advances by a small time step $dt$, we obtain the following expression for $E_{\text{thermomixer}}$:

$$E_{\text{thermomixer}} = h \times A \times (T_{\text{thermomixer}} - T(t))dt \qquad (5)$$

In Eq. 5, $h$ represents the heat transfer rate, $A$ the contact area, $T_{\text{thermomixer}}$ the temperature on the walls of the thermomixer (kept constant), and $T(t)$ the time-dependent temperature of the growth media inside the tube.

For $E_{\text{pumped in/out}}$, we obtain:

$$E_{\text{pumped in/out}} = m \times C_p \times M_{\text{frac,speed}}dt(T_{\text{in}} - T_{\text{out}}) \qquad (6)$$

Here $m$ denotes the mass of the growth media, $C_p$ its heat capacity (we used the heat capacity of water), and $M_{\text{frac,speed}}$ is the mass fraction circulation speed.

Finally, the change of energy in the tube can be written in terms of the temperature in the tube:

$$\Delta E_{\text{tube}} = m \times C_p \times dT(t) \qquad (7)$$

where $dT(t)$ represents the change in temperature resulting from the energy transfer.

In order to measure the heat transfer rate, we set the thermomixer temperature to 42 °C and the initial temperature of the growth media to the ambient temperature of the laboratory (25 °C). Moreover, we took into account the heat loss in the tubing connecting the incubation vessels. The complete experimental set-up is visually depicted in Fig. 2c, where each colored flask represents a thermomixer set at nominal temperatures of 25, 37, and 42 °C (yellow, light orange, and orange flasks, respectively), gray lines represent the tubing connecting the incubation vessels, and triangles within circles represent peristaltic pumps and the direction of the flow. We obtained the necessary additional parameters by measuring the length and the diameter of the tubing and the laboratory's ambient temperature. The heat transfer rates were 342.6975 and 82.3414 W m$^{-2}$ K$^{-1}$ for the thermomixers and the tubing, respectively. Solving the above equations we obtain the time-dependent temperatures ($T(t)$) for each of the three tubes. Our modeling reproduces the temperatures that we observed at the end of the actual experiments with dispersal with a mean error of 0.267 °C (Supplementary Table 9).

**Modeling growth at intermediate circulation speeds**. We modeled the growth of each strain with a lag phase followed by an exponential phase, with both phases depending on the strain and on temperature. To quantify the lag phase and the exponential growth phase for each of the strains at a given temperature, we use the following formula:

$$n(t) = n_0 \text{ for } t \le t_0 \qquad (8)$$

$$n(t) = n_0 \times \exp\left(r_{\text{growth}} \times (t - t_0)\right) \text{for } t > t_0 \qquad (9)$$

where $n(t)$ is the population density at time $t$, $n_0$ is the initial population density, $t_0$ is the length of the lag phase, and $r_{\text{growth}}$ is the constant exponential growth rate. We next grew the synthetic community at nominal temperatures of 25, 28, 34, 37, 40, and 42 °C without dispersal for 4 h (with the same starting population densities as in the experiments with dispersal and in three different falcon tubes—biological replicates—at each temperature) and we fit the observed population densities to the formula above while minimizing the SSE (Sum of Squared Errors) to obtain $t_0$ and $r_{\text{growth}}$. Our predicted population densities were in good agreement with the observed ones ($0.964 < R^2 < 1$, Supplementary Figure 12, Supplementary Table 10).

The population density $n(t)$ satisfies the equation:

$$dn(t)/dt = r(t) \times n \qquad (10)$$

where $r(t)$ is the growth rate, i.e., $r(t)$ is 0 during the lag phase and equal to $r_{\text{growth}}$ during the exponential phase.

To find the growth at the intermediate temperatures across the 25–40 °C temperature gradient that corresponds to the minimum and maximum observed temperatures during the experiments, we interpolated the calculated growth rates that we acquired from the growth assays of the mixed cultures without dispersal (i.e., at 25, 28, 34, 37, 40, and 42 °C—nominal temperature) using Piecewise Cubic Hermite Interpolating Polynomial (pchip). We used the growth rates of the mixed cultures because they include the interactions among the strains. We observed that strain B42 had an overall short lag phase (<30 min, Supplementary Figure 12) and a growth profile with two local maxima (at 31 and 36.5 °C) (Fig. 4b), that strain E310 had a longer lag phase than strain B42 (~60 min) and one global maximum (at 40 °C) (Fig. 4c), and that strain E111 had the longest lag phase (~ 120 min) and the sharpest growth maximum (at 37 °C) among the three (Fig. 4d).

For the scenarios with dispersal, we used the temperature model above to obtain the temperatures over time in all three vessels. We then added transport between the vessels according to the direction of the flow (Fig. 2c) and the circulation speed. Additionally, we subtracted a penalty term from the population density to account for the observed stress of the community due to dispersal (Supplementary Figure 6). We obtained the penalty term from the data as the best fitting parameter to the experimental data at the four different circulation speeds. This penalty manifested both as an increase in $t_0$ and as a decrease in $r_{\text{growth}}$ and was non-linear with increasing dispersal and strain-specific. For strain B42, the penalty affected mostly $r_{\text{growth}}$ and was up to −32% of $r_{\text{growth}}$ and up to +7.2 min for $t_0$. For strain E310, the penalty affected both $r_{\text{growth}}$ and $t_0$ and was up to −40.5% of $r_{\text{growth}}$ and up to +31 min for $t_0$. For strain E111, the penalty affected mostly $t_0$ and was up to −9% of $r_{\text{growth}}$ and up to +53.3 min for $t_0$.

For each strain, the dynamics at each vessel and each time point are modeled by:

$$dn_{25}(t)/dt = r \times (T_{25}(t), t) \times n_{25} + \text{disp}_{\text{speed}} \times (n_{42} - n_{25}) \qquad (11)$$

$$dn_{37}(t)/dt = r \times (T_{37}(t), t) \times n_{37} + \text{disp}_{\text{speed}} \times (n_{25} - n_{37}) \qquad (12)$$

$$dn_{42}(t)/dt = r \times (T_{42}(t), t) \times n_{42} + \text{disp}_{\text{speed}} \times (n_{37} - n_{42}) \qquad (13)$$

where $n_{25}(t)$, $n_{37}(t)$, and $n_{42}(t)$ are the population densities at time $t$ in the vessel with initial nominal temperature of 25, 37, and 42 °C, respectively, $r(T,t)$ is the interpolated growth rate with penalty subtracted at temperature $T$ and time $t$, and $\text{disp}_{\text{speed}}$ is the circulation speed. The second part of the equations gives the immigration ($I$) at each time point.

**Quantifying the homogeneity of the metacommunity**. To examine whether the metacommunity is homogenized at a given circulation speed (in the actual experiments and in the modeled scenarios of intermediate dispersal), we used the BC similarity. In specific, we compared for each dispersal rate the average pairwise BC within the metacommunity ("BC within") to the average pairwise BC among communities growing under a given degree of dispersal compared to the respective communities growing without dispersal at the same temperature ("BC across") (Fig. 3a).

For a given dispersal rate, "BC within" quantifies how similar the communities are within the metacommunity and "BC across" quantifies how much they change compared to when there is no dispersal. When the metacommunity is non-homogeneous, "BC across" should be significantly higher than "BC within", indicating that the communities did not change much compared to the zero dispersal conditions, whereas when the metacommunity is homogeneous "BC across" should be significantly lower than "BC within".

For the calculation of "BC within" and "BC across", we compared sample pairs from the same time point starting after the first hour of incubation. This was to avoid the artificial inflation of the BC indices; the populations of the three strains within the first hour of incubation are very close to their starting populations, irrespectively of the temperature (Supplementary Figure 12) or the circulation speed (Supplementary Figure 6) so the communities are always very similar during that time. Thus we calculated "BC within" and "BC across" based on 63 pairwise comparisons for the experimental data, i.e., three pairwise comparisons at seven time points for three independent replicate experiments per circulation speed, and based on 21 pairwise comparisons for the modeling data (three pairwise comparisons at seven time points). For both indices, the standard deviation increases as the mean similarity decreases (Fig. 5a, Supplementary Figures 8–9) because the communities become increasingly dissimilar within the metacommunity (for "BC within") and compared to when there is no dispersal (for "BC across") with increasing incubation time (Supplementary Figure 5). Communities from independent replicate experiments sampled at the same circulation speed, vessel, and time point were 96.2% (90.7–98.8%) similar on average (in terms of BC similarity).

**Calculating the strength of selection and immigration.** To examine the relative contribution of immigration and selection at the 100 scenarios with intermediate circulation speed, we computed the CV of the growth rates of each strain at different vessels, the average growth-over-immigration ratio in the metacommunity, and mean temperature difference among the vessels. For each circulation speed, we calculated the growth-over-immigration ratio at 10-min intervals for the whole period of 4 h before averaging. We calculated the CV of the growth rates of each strain and the mean temperature difference after the first 20 min of incubation when the temperature was stabilized (Fig. 4a). Finally, we examined the relative importance of immigration and selection by examining how the growth-over-immigration ratio, the CV of the growth rates of each strain, and the mean temperature difference change at the scenarios with intermediate circulation speeds.

**Additional simulations.** To further examine the importance of immigration and selection, we performed additional simulations under two scenarios where we excluded either the transfer of cells or the transfer of heat from vessel to vessel. Thus, in the scenario where the transfer of cells was excluded, the temperature of the vessels changed as in reality, according to the circulation speed (Fig. 4a). Likewise, in the scenario where the transfer of heat was excluded, the temperature of the vessels remained constant and cells were migrating according to the circulation speed. We examined the homogeneity of the metacommunity under both scenarios up to a circulation speed of 5.72 µl s$^{-1}$.

To further test the generality of our findings, we performed another two simulations where the dynamics of heat transfer, the range of circulation speed, and the growth of the strains were different compared to the actual experiment. In these simulations, we set the heat transfer rate from the thermomixers to the incubation vessels ~3 times lower than in the actual experiment. This caused the temperatures among the vessels to converge fast due to mixing for dispersal rates >4 µl s$^{-1}$ (Supplementary Figures 8A and 9A, upper left), because the thermomixers were not heating/cooling as fast as in the actual experiments. We further changed the range of circulation speed from 0 to 200 µl s$^{-1}$ (rather than from 0 to 71.5 µl s$^{-1}$ in the actual experiment). We also changed the growth profiles of the strains compared to what we observed in the actual experiment. In the first simulation, we gave a single global maximum to the growth of strain B42 at 35–35.5 °C, to the growth of strain E310 at 40 °C, and to the growth of strain E111 at 37 °C (Supplementary Figure 8A). We set the maximum of B42 higher than the maxima of the other two strains, followed by E310 and E111 (Supplementary Figure 8A). In the second simulation, we set the growth of strain B42 higher than the other two strains at a temperature range of 25–37 °C and the growth of strain E310 higher than that of E111 at all temperatures with a local maximum at 37.5 °C (Supplementary Figure 9A). Unlike the main model, here we applied a non-specific uniform penalty only to the exponential part of the growth of the strains. This penalty increased linearly with increasing dispersal at both simulations.

**Statistics.** For the comparison of "BC within" to "BC across", we used linear mixed-effects models regarding the experimental data and generalized mixed-effects models regarding the modeling data because the latter were not normally distributed (Anderson–Darling tests, two-sided $p > 0.1$ for all datasets). We set the BC to be the dependent variable "across/within" and "dispersal" to be the fixed effects and "time" and "vessel" to be the random effects. We corrected the $p$ values for multiple testing with the Bonferroni correction. The intercepts (random effects) were always very significant and there was no apparent autocorrelation of the residuals (Supplementary Figure 13). We performed the models in R[41] using the package lme4[42]. We performed the non-metric multidimensional (nMDS) analysis (for Supplementary Figure 5) in PRIMER v6[43] based on the BC similarity. We calculated the BC similarity for all analyses using the raw population densities (cells µl$^{-1}$) without any transformation, because these densities represent absolute abundance counts in communities where sampling is random, large, and fully covering the diversity. The large number of recorded events per sample, according to the Central Limit Theorem, allows us to safely estimate the mean population densities at a given time within an incubation vessel. That was also confirmed from the CV among the three technical replicates from each incubation vessel in the experiments with dispersal, which did not differ significantly from the expected variability caused by the instrument ($F$-test, $n = 108$, two-sided $p = 0.45$, compared against the instrument's CV of 3.75%). To examine potential interactions among the strains, we compared their growth rates in monocultures versus that in mixed cultures at 25, 37, and 42 °C using ANCOVA. In this analysis, we set the experimental time as the covariate and the log-transformed cell density as the dependent variable, and we examined the null hypothesis that the growth rates between monocultures and mixed cultures are equal.

**Reporting summary.** Further information on experimental design is available in the Nature Research Reporting Summary linked to this article.

## Data availability
Flow cytometric data are available in .fcs format online (http://flowrepository.org) under the "FR-FCM-ZYG6" and "FR-FCM-ZYTD" identifiers. The 16S rRNA gene sequences of the representative isolates have been deposited in the GenBank database under accession numbers MH998420–MH998449. Modeling code is available on GitHub (https://github.com/alexanderlorz/dispersal_vs_selection). The source data underlying Figs. 2d, e, 3b, 5a, and 6b, c are provided as a Source Data file.

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

## Acknowledgements

This publication is based upon work supported by the King Abdullah University of Science and Technology (KAUST) to D.D. under the baseline funding and the Office of Sponsored Research (OSR) Award No. OSR-2018-CARF-1973 to the Red Sea Research Center. The authors would like to thank Dr. Frederik Hammes and Dr. Jay Lennon for their constructive comments on a previous version of the manuscript, as well as Dr. Marco Fusi and Adam Bouchaala for their advice regarding statistical analyses and MATLAB coding, respectively.

## Author contributions

S.F. and A.L. conceived the original idea. S.F., A.L., and D.D. designed the study. S.F., A.L., A.V.-C. and A.B. performed the experiments. J.M.B., S.F. and A.V.-C. performed microscopy. A.L. and S.F. performed mathematical modeling. S.F., A.L. and A.V.-C. analyzed the data. S.F. and A.L. wrote the manuscript with input from all authors.

## Additional information

**Competing interests:** The authors declare no competing interests.

