## [Peer Review File · Nature Communications]

Editorial Note: This manuscript has been previously reviewed at another journal that is not operating a transparent peer review scheme. This document only contains reviewer comments and rebuttal letters for versions considered at *Nature Communications*. Mentions of the other journal and previous referee reports have been redacted.

Reviewers' comments:

Reviewer #1 (Remarks to the Author):

The authors have done a thorough job of addressing reviewer comments, providing new data and analyses. This work will make a nice contribution to microbial ecology. Future work is needed to truly separate potential selection from immigration (e.g. what if temperature isn't the variable that is being homogenized by dispersal? what if it's some other component in the medium?). However, this work is important and interesting as is.

My only comment is that the last paragraph of the discussion should be reorganized. As it stands, the paper ends by highlighting reviewer criticisms and outlining future work that should be done. While this is important to include at the end of the discussion, the authors should end with a sentence or two that highlights the overall importance of their results and their ramifications.

Reviewer #2 (Remarks to the Author):

The authors have addressed all my concerns. I do not have any further comments.

Reviewer #3 (Remarks to the Author):

Fodelianakis et al. describe the use of a synthetic microbial community to evaluate the relative importance of immigration to homogenization of microbial communities. I highly commend the authors on their rigor in addressing most of my major (technical) concerns in their revised manuscript. I believe the quality has significantly improved and I have become more convinced on the data presented. The overall statistical analysis of the experimental work is now thorough and rigorous enough to substantiate the findings. All data is now publicly available as well. However, I still have some constructive comments on the current manuscript.

I believe the authors have some misconceptions on the method I proposed by Rubbens et al to replace their manual gating strategy, which are reflected on lines 317 - 325. The method by Rubbens et al, does not necessitate that cultures are in stationary growth phase or that they are stained with a nucleic acid stain. The method merely replaces the manual gating on 2 dimensions by a random forest model that creates a gating template on an arbitrary user-specified number of parameters (for example FSC/SSC). It is trained on the individual culture data and then applied on mixed culture data. Given its "supervised" nature, it should always perform better than manual gating. Please adjust this part of the text.

However, and obviously, if manual gating suffices there is no need to delve into this approach. But when I look at the available supplementary information I observe "spill-over" values or as I understand, % of misclassification, of ~ 6 - 15% between E310 and B42 populations, and > 10% between the E111 and E310 populations (even 43.2% at 42°C for the E111-E310 populations!). The authors simply used these spill-over values as correcting factors to calculate the individual population abundances at the individual time points. However I'm not a proponent of such an approach, as it makes some strong assumptions: i.e., the growth curve, and the physiology of the population remains the same during co-culturing in a mixed community, as during the experiment in which the spill-over factor was determined. Given that strong interactions between these populations were determined, these will have an effect on the physiology and size distribution of your individual populations. At the bare minimum the authors should comment on and acknowledge that the biological variability (just look at the variability in growth curves of the axenic cultures in Fig S4!!) due to interactions will result in the (unavoidable) misclassification of cells, which is a limitation of the method used.

Lastly, the title of the manuscript ("Dispersal homogenizes communities via immigration even at low rates") suggests (to me) that the findings of the manuscript are generalizable to all (microbial?) communities. However, only a single three-member synthetic community was tested consisting of phylogenetically distinct taxa. This study concerns 3 taxa isolated from a desert soil environments, which brings with it certain limitations on what can be inferred from this experiment. For example, in the revised manuscript, the authors show that there is strong evidence for interactions between these three populations. As such, the level of interaction between the partners may strongly mediate the relative effect of dispersal, and thus different synthetic communities may yield different results in function of the level of interaction. Although, one cannot control for all variables and extrapolate to all possible conditions, I feel that some of the limitations of their findings and experimental design, which the authors excellently describe in the discussion (line 326 and onwards) should be clear from the title. Nonetheless, the experimental design and the clever use of quantitative flow cytometry is a clear way forward for future microbial ecology studies.

Line 200: please specify what you mean with "in silico"

Reviewer #5 (Remarks to the Author):

[Redacted]

I appreciate the authors' detailed response and efforts in conducting additional experiments and analysis. However, these efforts have not assuaged my two main concerns regarding the evidence for the presence of interspecies interactions, and for migration being the dominant homogenizing mechanism.

Regarding the interactions between species, I find it extremely surprising that three species that were not specifically selected to interact positively would form mutualisms. This is far from my experience working with soil isolates, and mutualisms have been shown to be quite rare among bacterial isolates (e.g. Foster, Kevin R., and Thomas Bell. "Competition, not cooperation, dominates interactions among culturable microbial species." *Current biology* 22.19 (2012): 1845-1850.) This result may be an artefact stemming from the fact that the monoculture and mixed cultures experiments that were compared in order to infer interactions were started from different initial densities, with the mixed cultures starting from a density ~5-10 times lower than monocultures (according to Supplementary Dataset 1). Monocultures may have grown by a smaller factor not due to interactions, but due to slower growth at their higher population densities. Such slower growth at higher densities is apparent for E111 at 37C and 42C (it's hard to gauge it by eye on a linear scale for the other species and temperatures). Additionally, it's not clear to me whether the monoculture and mixed culture experiments were conducted at the same time. If they were done at different time, and one time each, than a quantitative comparison may be misleading, even if there are biological replicates. This is because there can be systematic differences between the conditions on different days (e.g. different batches of LB) leading to systematic quantitative differences in growth parameters.

Regarding the support for migration being the dominant homogenizing mechanism, the new analysis looking at growth rate differences (Fig 5B) is interesting and consistent with the authors' claim. However, I still feel like there are alternative explanations, and stronger evidence is needed. For example, one alternative explanation for the observation is higher circulation rates leading to slower growth overall - which seems to be the case in Fig. S6 (though a log-scale would make this easier to evaluate by eye), and the circulation-dependent penalty on growth rates needed to fit the data. Slower growth (or longer lag) would mean that communities that start out similar simply diverge more slowly, rather than 'become homogenized', resulting in high 'BC within'. The low 'BC

across' is due to communities at high circulation rates remaining more similar to their initial composition, whereas the faster-growing communities without circulation diverge from it. As I suggested in my original review, this can be investigated using simulations in which cells migrate but temperature is not affected, and ones in which circulation affects the temperatures without migration of cells. Additional factors, such as the circulation-dependent growth penalty, can be modulated in these simulations to identify specific plausible mechanisms affecting the similarity between communities.

Finally, I still feel like there is a need for direct experimental evidence, where communities are coupled either only thermally, or solely by migration. As an experimental microbial ecologist, I fully appreciate how challenging this can be and that my previous suggestions for specific experimental setups are naive at best, but I believe that direct experimental evidence should be the standard for laboratory-based model systems.

Additional comments

In the model, each species grows exponentially with a constant growth rate, but one that's different in monoculture and in a mixed community. This is a rather unusual modeling choice, as interactions typically depend on species densities, rather than their on their presence or absence. It may be a justified choice for the conditions considered here, but the choice and its rationale should be clarified to readers. Relatedly, the growth penalty term due to circulation is still not clear to me. Is the lag phase affected at all? What are the actual fitted penalty values? Does Supplementary Dataset 2 list the rates with the penalty subtracted or without? For readers to be able to reproduce the authors' work, more details regarding this should be provided in the manuscript or supplement (including the ones provided in the authors' reviewer response).

In supplementary dataset 1, the time points are not independent, and the p-value would therefore be inflated. The initial cell densities reported in supplementary dataset 1 and legend of Fig. S4 are inconsistent.

Please note that the reviewers' comments are in plain text our responses are in bold italics. The line numbers correspond to the clean updated version of our manuscript and not to the manuscript where changes are tracked.

Reviewers' comments:

Reviewer #1 (Remarks to the Author):

The authors have done a thorough job of addressing reviewer comments, providing new data and analyses. This work will make a nice contribution to microbial ecology. Future work is needed to truly separate potential selection from immigration (e.g. what if temperature isn't the variable that is being homogenized by dispersal? what if it's some other component in the medium?). However, this work is important and interesting as is.

We thank the reviewer for the positive assessment.

My only comment is that the last paragraph of the discussion should be reorganized. As it stands, the paper ends by highlighting reviewer criticisms and outlining future work that should be done. While this is important to include at the end of the discussion, the authors should end with a sentence or two that highlights the overall importance of their results and their ramifications.

We agree with the reviewer, and we are now ending the discussion highlighting the overall importance of our study (Lines 380-383).

Reviewer #2 (Remarks to the Author):

The authors have addressed all my concerns. I do not have any further comments.

We thank the reviewer for the positive assessment.

Reviewer #3 (Remarks to the Author):

Fodelianakis et al. describe the use of a synthetic microbial community to evaluate the relative importance of immigration to homogenization of microbial communities. I highly commend the authors on their rigor in addressing most of my major (technical) concerns in their revised manuscript. I believe the quality has significantly improved and I have become more convinced on the data presented. The overall statistical analysis of the experimental work is now thorough and rigorous enough to substantiate the findings. All data is now publicly available as well. However, I still have some constructive comments on the current manuscript.

We thank the reviewer for the positive assessment and for the additional helpful comments. We have now addressed all these remaining comments in the revised version of the manuscript. Please see our responses below.

I believe the authors have some misconceptions on the method I proposed by Rubbens et al to replace their manual gating strategy, which are reflected on lines 317 - 325. The method by Rubbens et al, does not necessitate that cultures are in stationary growth phase or that they are stained with a nucleic acid stain. The method merely replaces the manual gating on 2 dimensions by a random forest model that creates a gating template on an arbitrary user-specified number of parameters (for example FSC/SSC). It is trained on the individual culture data and then applied on mixed culture data. Given its “supervised” nature, it should always perform better than manual gating. Please adjust this part of the text.

We thank the reviewer for the clarification. We were referring to the discussion of the cited manuscript where the authors mention the limitations of that specific work (in their manuscript, cultures were indeed stained, and screened only at the stationary phase), whereas the reviewer is referring to the random forest algorithm that is indeed more general and can be applied on any two signal axes. We have now updated the respective part in the discussion (Lines 345-353), also adding comments on the reviewer’s observation below (see below for more details).

However, and obviously, if manual gating suffices there is no need to delve into this approach. But when I look at the available supplementary information I observe “spill-over” values or as I understand, % of misclassification, of ~ 6 - 15% between E310 and B42 populations, and > 10% between the E111 and E310 populations (even 43.2% at 42°C for the E111-E310 populations!). The authors simply used these spill-over values as correcting factors to calculate the individual population abundances at the individual time points. However I’m not a proponent of such an approach, as it makes some strong assumptions: i.e., the growth curve, and the physiology of the population remains the same during co-culturing in a mixed community, as during the experiment in which the spill-over factor was determined. Given that strong interactions between these populations were determined, these will have an effect on the physiology and size distribution of your individual populations. At the bare minimum the authors should comment on and acknowledge that the biological variability (just look at the variability in growth curves of the axenic cultures in Fig S4!!) due to interactions will result in the (unavoidable) misclassification of cells, which is a limitation of the method used.

We agree with the reviewer regarding the potential changes in the physiology of the cells in the mixed cultures compared to the pure cultures and that this could lead to significant misclassification of cells. However, we have evidence from two independent methods that this did not happen in our case. First, we have performed extensive screening of pure and mixed cultures with scanning electron microscopy at various growth stages (we present some photos at Figure S1) and we have not witnessed any significant differences between pure and mixed cultures. Second, when we screened the mixed cultures with flow cytometry we could always observe the individual populations at the positions where they were supposed to be based on the profiles of the pure cultures (see for example

the mixed culture profile in Fig. 2B bottom, or any mixed culture sample in the FlowRepository dataset). This indicates that the scattering profiles of each strain, and thus the spillover ratios, were preserved in the mixed cultures. Moreover, because we use ratios to correct for spillover among the manual gates, changes in the growth rates at different temperatures do not affect the classification accuracy of the individual populations. In our work, the selection of strains with very distinct scattering profiles at all stages of the 4-hour growth assay served as an “internal control” to verify the validity of our gating strategy; we could always observe the individual populations. In other works where this may not be possible, the use of the random forest model of Rubbens et al. will be very useful to correctly classify the observed events into the correct populations. We now discuss this in the revised version of the manuscript (Lines 345-353).

Lastly, the title of the manuscript (“Dispersal homogenizes communities via immigration even at low rates”) suggests (to me) that the findings of the manuscript are generalizable to all (microbial?) communities. However, only a single three-member synthetic community was tested consisting of phylogenetically distinct taxa. This study concerns 3 taxa isolated from a desert soil environments, which brings with it certain limitations on what can be inferred from this experiment. For example, in the revised manuscript, the authors show that there is strong evidence for interactions between these three populations. As such, the level of interaction between the partners may strongly mediate the relative effect of dispersal, and thus different synthetic communities may yield different results in function of the level of interaction. Although, one cannot control for all variables and extrapolate to all possible conditions, I feel that some of the limitations of their findings and experimental design, which the authors excellently describe in the discussion (line 326 and onwards) should be clear from the title. Nonetheless, the experimental design and the clever use of quantitative flow cytometry is a clear way forward for future microbial ecology studies.

We thank the reviewer for the comment. We agree with the reviewer regarding the generalization of our findings, even though we tried to generalize our study as much as possible by performing additional simulations where we changed the growth rates of the strains (and thus their interactions) (Fig. S8-S9, Lines 282-294). We have now changed the title of the manuscript to “Dispersal homogenizes communities via immigration even at low rates in a simplified synthetic bacterial metacommunity” to make it more specific to the readers.

Line 200: please specify what you mean with “in silico”

Done. We replaced “in silico” with “resulting modeled”, referring to the communities resulting from the model output (Line 201).

Reviewer #5 (Remarks to the Author):

[Redacted]

I appreciate the authors' detailed response and efforts in conducting additional experiments and analysis. However, these efforts have not assuaged my two main concerns regarding the evidence for the presence of interspecies interactions, and for migration being the dominant homogenizing mechanism. ***We have now performed additional experiments (growth assays of pure strains starting from lower cell densities) and analyses (more simulations and ANCOVAs to replace χ^2 tests), and we are confident that we provide ample evidence for the existence of interspecies interactions and for the dominant role of migration in homogenizing the metacommunity. Please see our detailed responses below.***

Regarding the interactions between species, I find it extremely surprising that three species that were not specifically selected to interact positively would form mutualisms. This is far from my experience working with soil isolates, and mutualisms have been shown to be quite rare among bacterial isolates (e.g. Foster, Kevin R., and Thomas Bell. "Competition, not cooperation, dominates interactions among culturable microbial species." *Current biology* 22.19 (2012): 1845-1850.) This result may be an artefact stemming from the fact that the monoculture and mixed cultures experiments that were compared in order to infer interactions were started from different initial densities, with the mixed cultures starting from a density ~5-10 times lower than monocultures (according to Supplementary Dataset 1). Monocultures may have grown by a smaller factor not due to interactions, but due to slower growth at their higher population densities. Such slower growth at higher densities is apparent for E111 at 37C and 42C (it's hard to gauge it by eye on a linear scale for the other species and temperatures). Additionally, it's not clear to me whether the monoculture and mixed culture experiments were conducted at the same time. If they were done at different time, and one time each, than a quantitative comparison may be misleading, even if there are biological replicates. This is because there can be systematic differences between the conditions on different days (e.g. different batches of LB) leading to systematic quantitative differences in growth parameters.

We agree with the reviewer that mutualism is generally less frequent than antagonism in soil isolates. However, we have performed additional growth assays and we show that in our case there are indeed mutualistic relationships among the three strains that we used and these relationships occur in the same way both at high and at low starting population densities (please see the revised Supplementary Dataset 1). In our original growth assays, our aim was that the total starting cell density was comparable between the monocultures and the mixed cultures and this resulted in maximum 2-3 times higher starting densities in the former compared to the latter. In our additional pure culture growth assays, we started from cell

densities that are comparable to the starting densities of the individual populations in the mixed cultures as suggested by the reviewer. These additional assays yielded qualitatively identical results to our original ones, showing that the growth rates of the strains in monoculture are significantly lower than the respective growth rates in mixed cultures at all temperatures (except where the comparison is not feasible due to non-linearity in the increase of the log-transformed cell density with time, please see the revised Supplementary Dataset 1). We also revised the way that we compare these growth rates according to the additional comment of the reviewer, using analysis of covariance (ANCOVA) instead of the originally used χ^2 tests. All data and comparisons, both at low and at high starting cell densities of monocultures, are now included in the revised Supplementary Dataset 1 and the rationale of the comparisons is explained in Materials & Methods (Lines 485-490).

The original single and mixed growth assays were performed on consecutive days and the growth medium was prepared once and used in all experiments to avoid potential biases. Of course, the additional assays in the revised version were performed several months later and with a different batch of growth medium. However, the overall high reproducibility of the results suggests that there are no significant systematic differences among our assays. The raw flow cytometry data from the additional growth assays of the pure strains have been deposited in FlowRepository under the ID "FR-FCM-ZYTD" and can be accessed by the reviewers at the following link:

<http://flowrepository.org/id/RvFrmfc0efQijVkxPdm2GOoOwgSfy5YQayKOgPMZy6ze8MNL1euqOU4ZlbSAGErz>

All raw data will be made available upon potential acceptance of the manuscript.

Regarding the support for migration being the dominant homogenizing mechanism, the new analysis looking at growth rate differences (Fig 5B) is interesting and consistent with the authors' claim. However, I still feel like there are alternative explanations, and stronger evidence is needed. For example, one alternative explanation for the observation is higher circulation rates leading to slower growth overall - which seems to be the case in Fig. S6 (though a log-scale would make this easier to evaluate by eye), and the circulation-dependent penalty on growth rates needed to fit the data. Slower growth (or longer lag) would mean that communities that start out similar simply diverge more slowly, rather than 'become homogenized', resulting in high 'BC within'. The low 'BC across' is due to communities at high circulation rates remaining more similar to their initial composition, whereas the faster-growing communities without circulation diverge from it.

We agree with the reviewer that slower growth at high dispersal speeds could artificially affect 'BC within' and 'BC across'. However, this did not happen in our study because the communities were diverging rapidly from the initial communities at all circulation speeds except perhaps at the

highest speed ($71.5 \mu\text{l sec}^{-1}$). This is evident from Figure S5; if the communities were not diverging from the initial, the triangles should all be clustered together. This might be the case only for the highest speed (Fig. S5D) but it is definitely not the case for the other experiments and especially for the experiments at $5 \mu\text{l sec}^{-1}$ where the metacommunity is already homogenized (Fig. S5B). To further demonstrate this fact, we have performed an additional analysis where we compare the decay in the BC similarity at each vessel through time, between the experiments at $1.75 \mu\text{l sec}^{-1}$ (where the metacommunity is still non-homogeneous) and $5 \mu\text{l sec}^{-1}$ (where the metacommunity is homogeneous) (see Fig. R1_1 below). We show, using analyses of covariance where we compare the slopes of the decay, that the decay rate was similar at both dispersal speeds except from the vessels at 25°C where the decay rate at $5 \mu\text{l sec}^{-1}$ was actually slightly faster than at $1.75 \mu\text{l sec}^{-1}$. Since this information is already included in Figure S5, we prefer not to include this additional analysis in the main text or in the Supplementary Information. Readers that are potentially interested in further details can refer to the reviewers' comments and responses that will be available upon potential acceptance of the manuscript in Nature Communications, as per the journal's editorial policy.

Figure R1_1. Comparisons of the decay in the Bray-Curtis similarity (% , y-axis) with increasing incubation time (pair-wise time difference compared to the communities at the beginning of the experiment – x-axis) between the experiments with circulation speeds of 1.75 and 5 $\mu\text{l sec}^{-1}$ at each incubation vessel. The temperature (nominal) of each vessel is written on top of each graph. The statistics within each plot concern ANCOVA results, comparing the decay rates between the experiments at different circulation speeds. Black and red stands for samples from experiments at 1.75 and 5 $\mu\text{l sec}^{-1}$, respectively.

As I suggested in my original review, this can be investigated using simulations in which cells migrate but temperature is not affected, and ones in which circulation affects the temperatures without migration of cells. Additional factors, such as the circulation-dependent growth penalty, can be modulated in these simulations to identify specific plausible mechanisms affecting the similarity between communities.

We thank the reviewer for the excellent suggestion; it was not clear to us in his/her original review. We have now performed such additional simulations in which a) temperature changes as if there was dispersal but cells do not migrate from vessel to vessel, b) cells migrate but temperature remains constant at the initial values. We examine changes in “BC within” and “BC across” under both of these scenarios in a gradient of circulation speeds from 0.05 to 5.72 $\mu\text{l sec}^{-1}$ where the metacommunity is already homogenized in reality (it is already homogenized in the experiments at 5 $\mu\text{l sec}^{-1}$, Figure 3B). We show that the metacommunity is homogenized very similarly as in the original model under scenario b) but not under scenario a) where ‘BC within’ and ‘BC across’ do not even converge (Lines 257-270, Figure 6). We thus now provide additional evidence suggesting that immigration, not thermal coupling, is responsible for the homogenization of the metacommunity. Temperature homogenization definitely contributes more at higher circulation speeds, as observed by the convergence in the growth rates of the strains (Fig. 5B). Regarding the effect of the penalty, we have already performed simulations where the nature of the penalty was different than in our main model (Lines 282-294, 726-728, Fig. S8-S9) and we obtained qualitatively similar results to our main model.

Finally, I still feel like there is a need for direct experimental evidence, where communities are coupled either only thermally, or solely by migration. As an experimental microbial ecologist, I fully appreciate how challenging this can be and that my previous suggestions for specific experimental setups are naive at best, but I believe that direct experimental evidence should be the standard for laboratory-based model systems.

We agree with the reviewer regarding the value of direct experimental evidence. However, as we have stated in our original responses, it is nearly impossible to replicate the given experimental setup with high fidelity while eliminating either migration or temperature convergence. Nevertheless, we made some additional efforts to artificially converge temperatures among

three vessels originally at (nominal) 25, 37 and 42°C (something similar to scenario b of our additional simulations described above). We tried to program the temperatures of the thermomixers at 37 and 42°C to drop, and the thermomixer at 25°C to increase, after a given time in order to mimic the temperature dynamics during the actual experiments with circulation (Fig. 4A). We made quite a few trials but we were unable to replicate the temperature dynamics closely enough to guarantee that a potential experiment would not be biased by the differences between the experimental setups.

We believe that in the revised version of our manuscript we provide ample evidence to support our claim that the metacommunity is homogenized by immigration. We clearly state the limitations of our current methodology and analyses in the Discussion, and we think that it is up to the reader and up to future researchers to critically evaluate and improve our findings, respectively.

Additional comments

In the model, each species grows exponentially with a constant growth rate, but one that's different in monoculture and in a mixed community. This is a rather unusual modeling choice, as interactions typically depend on species densities, rather than their on their presence or absence. It may be a justified choice for the conditions considered here, but the choice and its rationale should be clarified to readers.

The growth rates of the single strains are not relevant for our model, because in our model there are always three strains. Thus, we only take into account the growth rates in the mixed community that depend on species interactions. The growth of the single strains was only used initially to demonstrate that there is strong selection, per strain, across the examined temperature gradient and that there are important interactions among the strains. We now clarify the use of the mixed culture growth rates in Materials & Methods (Lines 622-623).

Relatedly, the growth penalty term due to circulation is still not clear to me. Is the lag phase affected at all? What are the actual fitted penalty values? Does Supplementary Dataset 2 list the rates with the penalty subtracted or without? For readers to be able to reproduce the authors' work, more details regarding this should be provided in the manuscript or supplement (including the ones provided in the authors' reviewer response).

We agree with the reviewer that the nature of the penalty was not fully explicit in the previous version of our manuscript. Yes, in our modeling the lag phase (t_0) is also affected by the increasing circulation speed; the penalty is applied to both the lag phase and to the exponential part of the growth (r_{growth}). The penalty is non-linear and strain-specific, and it affects differently the t_0 and r_{growth} of each strain. For strain B42 the penalty affected mostly r_{growth} and was up to -32% of r_{growth} and up to +7.2 min for

t0. For strain E310 the penalty affected both r_{growth} and $t0$ and was up to -40.5% of r_{growth} and up to +31 min for $t0$. For strain E111 the penalty affected mostly $t0$ and was up to -9% of r_{growth} and up to +53.3 min for $t0$. We have now added these details regarding the nature and the range of the penalty in Materials & Methods (Lines 635-640). Supplementary Dataset 2 lists the rates without the penalty subtracted. Any reader that would like to reproduce the model can do so by running the MATLAB code that we provide online (<http://alexanderlorz.com/dispersal-dominates-over-selection.html>). This code also includes a raw data file that contains all the lag phase lengths and exponential growth rates at different dispersal rates ('KKX0.mat' file). The data in this file are based on our experimental setup and on the specific strains, but the user can modify the values (e.g. after performing experiments with different strains) and run the code to test different scenarios.

In supplementary dataset 1, the time points are not independent, and the p-value would therefore be inflated.

We thank the reviewer for the observation. We have now substituted the χ^2 tests with more appropriate ANCOVAs to account for this fact (Supplementary Dataset 1).

The initial cell densities reported in supplementary dataset 1 and legend of Fig. S4 are inconsistent.

We thank the reviewer for noticing such inconsistency. Indeed the cell densities in the legend of Figure S4 were incorrect. We were erroneously reporting the values within the gates and not the total cells (i.e., the gate counts divided by the respective percentages in Table S3). We have corrected this in the revised version of the manuscript (i.e., in the legend of Figure S4).

REVIEWERS' COMMENTS:

Reviewer #3 (Remarks to the Author):

The authors have addressed all my concerns. I commend the authors on this interdisciplinary study, and hope to encounter follow-up studies using more complex synthetic communities in the (near) future!

Reviewer #5 (Remarks to the Author):

I thank the authors for their detailed response, which has addressed all my comments.